# SelfMask: Cross-modal Self-Masking for Multimodal Representation Learning in Missing Modality Scenarios

## Abstract

Multimodal learning promises to harness complementary information across diverse modalities, yet real-world deployments often face missing modalities due to acquisition costs, privacy constraints, or data corruption, leading to substantial performance degradation. We present SelfMask, a framework for learning robust representations in the presence of incomplete multimodal data. During training, SelfMask imputes missing modality representations through a masked representation learning scheme with adaptive masking, where informative masks are learned from data rather than sampled at random. To guide the imputation without relying on unavailable ground-truth for missing modalities, we introduce a cross-modal consistency loss: predicted representations of missing modalities are required not only to align with semantic content but also to support the reconstruction of observed ones. This consistency-based objective encourages robust, semantically grounded representations. Experiments on MIMIC-IV and CMU-MOSEI demonstrate that SelfMask consistently improves resilience and predictive accuracy under diverse missing-modality scenarios. Ablation studies further show that our learned masks outperform conventional random masking, yielding more reliable cross-modal representations. Our framework is broadly applicable across multimodal domains, offering a practical solution for real-world settings where incomplete modalities are the norm.

## 1 Introduction

Multimodal learning has emerged as a powerful paradigm for integrating complementary information from heterogeneous sources, such as images, text, and time-series signals (Wu et al., 2024; Zong et al., 2024). By exploiting cross-modal interactions, multimodal approaches often yield richer representations and superior predictive performance compared to unimodal methods (Wu et al., 2024). Yet, most existing frameworks rely on the strong assumption that all modalities are simultaneously available during both training and inference. In practice, however, this assumption is rarely satisfied. Modalities may be absent due to acquisition costs, privacy restrictions, sensor malfunction, or data corruption (Hayat et al., 2022; Zhang et al., 2023; Yao et al., 2024). For instance, many patients' medical records lack imaging scans, while online media data may include text but not audio. Such incomplete multimodal conditions often lead to severe performance degradation, undermining the reliability of multimodal systems in real-world applications (Ma et al., 2021; Li et al., 2025).

Existing approaches to missing modality scenarios can be categorized into three main paradigms. *Generative imputation methods* (Boyko et al., 2025; Yao et al., 2024) reconstruct missing raw data through autoencoder architectures, but they may be expensive for high-dimensional modalities and exhibit semantic drift in generated content that degrades downstream task performance. *Feature alignment approaches* (Wang et al., 2023; Zhang et al., 2023) project modalities into shared representation spaces using linear or nonlinear transformations, assuming modalities contain overlapping semantic information—an assumption that fails for modalities with disjoint information content or different temporal resolutions. *Domain-specific architectures* (Hayat et al., 2022; Xu et al., 2024; Li et al., 2025) incorporate task-specific inductive biases through specialized attention mechanisms or fusion layers, limiting their applicability to new domains without architectural modifications.

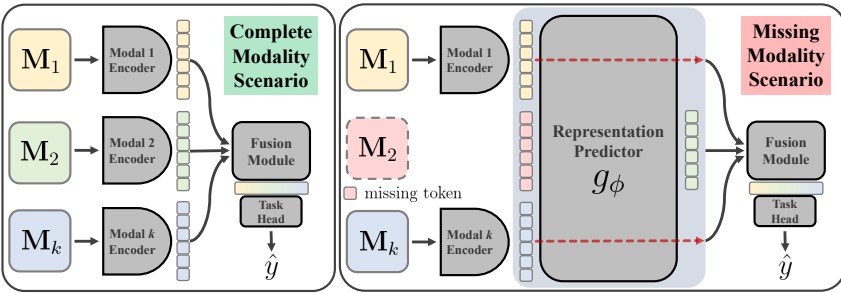

Figure 1: **Missing modality scenario on multimodal data.**

While the three paradigms above cover the dominant directions, existing methods also adopt related strategies that attempt to refine or extend them. Meta-learning approaches (Ma et al., 2021; Zhang et al., 2023) require extensive support sets with complete modality pairs during training, limiting their applicability to scenarios with sparse supervision. Disentanglement-based methods (Wang et al., 2023; Yao et al., 2024; Xu et al., 2024; Li et al., 2025) optimize modality-specific reconstruction objectives that fail to generalize across heterogeneous data distributions. Masked autoencoder variants (Boyko et al., 2025; Shah et al., 2023) achieve reconstruction fidelity through computationally expensive generative modeling in high-dimensional input spaces, while contrastive methods depend critically on carefully designed positive-negative pair sampling strategies. These approaches fundamentally treat missing modalities as a data corruption problem rather than leveraging modality incompleteness as a regularization mechanism for learning robust cross-modal representations.

In this work, we introduce SELFMASK, a learning framework designed to handle multimodal data with missing modalities. Inspired by masked representation learning frameworks (He et al., 2022; Assran et al., 2023), our framework reconstructs the representations of missing modalities from masked inputs. However, unlike conventional masked representation learning that relies on randomly sampled masks, we propose an *adaptive masking strategy* that learns informative masking patterns directly from data, thereby improving the efficiency of multimodal representation learning. To further enhance cross-modal robustness, we optimize a representation matching loss that explicitly emphasizes consistency across modalities. Although our method can be viewed as an imputation approach, it differs fundamentally from prior imputation methods that operate in raw input space: instead, we directly impute missing modality representations, allowing the model to focus on capturing and recovering core semantic content rather than superficial details. We evaluate our method on the MIMIC-IV and CMU-MOSEI benchmarks, demonstrating consistent gains in both resilience and predictive accuracy across a range of missing-modality scenarios.

## 2 METHODS

### 2.1 PROBLEM SETUP AND NOTATIONS

Consider a multimodal dataset with $M$ modalities with $N$ data points. Each data point may have different subsets of modalities observed, and we denote $\mathcal{O}_i \subseteq [M] := \{1, \dots, M\}$ the set of *observed* modalities and $\mathcal{M}_i := [M] \setminus \mathcal{O}_i$ the set of *missing* modalities for the $i^{\text{th}}$ data point. The dataset $\mathcal{D} = \{(\mathbf{X}_i^{(\mathcal{O}_i)}, y_i)\}_{i=1}^N$ consists of a pair of inputs $\mathbf{X}_i^{(\mathcal{O}_i)} := (\mathbf{X}_i^{(m)})_{i \in \mathcal{O}_i}$ with $\mathbf{X}_i^{(m)}$ denoting the $i^{\text{th}}$ input from $m^{\text{th}}$ modality, and $y_i$ is a label. Similar to $\mathbf{X}_i^{(\mathcal{O}_i)}$, we denote the input for the missing modality as $\mathbf{X}_i^{(\mathcal{M}_i)}$. For notational simplicity, we omit the sample index $i$ when the context is clear.

Our primary objective is to learn a robust predictor that maps an input with possibly missing modalities to a quantity to predict. A general framework to achieve this consists of modality-specific encoders and a subsequent fusion module. We denote the encoder for the $m^{\text{th}}$ modality as $\mathcal{E}^{(m)}$ which maps an input $\mathbf{X}^{(m)}$ to a sequence of latent representations $\mathbf{Z}^{(m)} = [\mathbf{z}_1^{(m)}, \dots, \mathbf{z}_{T^{(m)}}^{(m)}] \in \mathbb{R}^{T^{(m)} \times d}$, where $T^{(m)}$ is the sequence length and $d$ is the dimension of the representations. These sequences naturally arise from different data structures: image patches in Vision Transformers (ViTs) (Dosovitskiy et al., 2020), token embeddings in text, or temporal segment embeddings in time-series data.

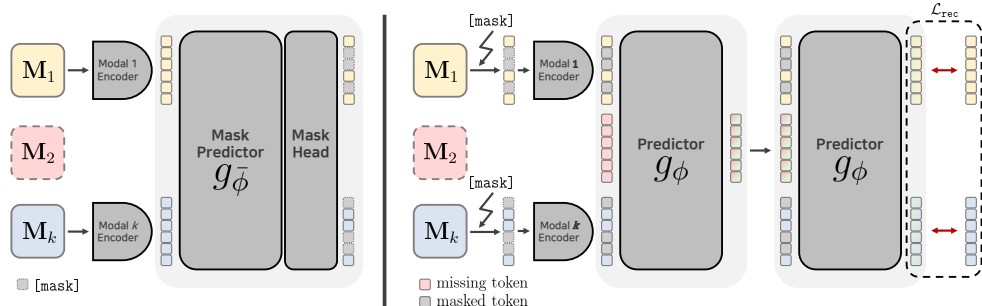

Figure 2: **Overview of SELFMASK.** The illustration emphasizes how cross-modal self-masking provides auxiliary supervision for observed modalities while guiding the representation predictor to impute coherent missing-modality embeddings, ultimately supporting robust multimodal fusion.

The representations are aggregated by a fusion module $f_\psi$ with learnable parameters $\psi$ to produce the final prediction $\hat{y}$.

When some modalities are missing, multimodal models must handle incomplete inputs $\mathbf{x}^{(m)} m \in \mathcal{O}$ while still producing reliable predictions. A common strategy is to impute missing values with trivial substitutes (e.g., zeros or learnable parameters), but this approach fails to capture important cross-modal relationships that influence the final prediction. To address this, we propose learning a *representation predictor* $g_\phi$ with parameters $\phi$, which takes encoded representations from observed modalities and imputes the missing ones by leveraging cross-modal interactions. To achieve this, we introduce SELFMASK, a simple yet effective cross-modal learning framework for training the representation predictor. An overview of the framework is shown in Figure 2.

## 2.2 MISSING MODALITY REPRESENTATION IMPUTATION

To learn the predictor $g_\phi$, we adopt a masked representation learning approach that learns the model by reconstructing the representations from masked inputs (He et al., 2022). Instead of predicting raw signals, we learn the predictor to reconstruct in representation space, thereby avoiding wasted capacity on reconstructing minor artifacts or low-level details in high-dimensional spaces and encouraging it to focus on capturing semantic content (Assran et al., 2023; Bardes et al., 2025; Assran et al., 2025).

The overall training pipeline goes as follows. Given an input $\mathbf{x}^{(\mathcal{O})}$, we first apply the masks $\mathbf{m}^{(m)}$ for each modality $\mathbf{x}^{(m)}$ in the observed modality set $\mathcal{O}$. For instance, if the $m^{\text{th}}$ modality were images, the masks may be a binary matrix that mask out some pixels or patches of the images. Then we put the masked input to the encoder $\mathcal{E}^{(m)}$ to get

$$\tilde{\mathbf{Z}}^{(m)} := [\tilde{\mathbf{z}}_1^{(m)}, \ldots, \tilde{\mathbf{z}}_{T^{(m)}}^{(m)}] = \mathcal{E}^{(m)}(\mathbf{m}^{(m)} \odot \mathbf{x}^{(m)}) \in \mathbb{R}^{T^{(m)} \times d}. \quad (1)$$

For missing modality $m \in \mathcal{M}$, we introduce a learnable missing token $\mathbf{t}^{(m)} \in \mathbb{R}^d$ that is replicated to form $\mathbf{T}^{(m)} = \mathbf{1}_{T^{(m)}} \otimes \mathbf{t}^{(m)} \in \mathbb{R}^{T^{(m)} \times d}$, where $\mathbf{1}_{T^{(m)}}$ is a vector of ones of length $T^{(m)}$. Finally, before putting the representations into the predictor $g_\phi$, we add learnable modality embeddings $\mathbf{e}^{(m)} \in \mathbb{R}^d$ as replicated form of $\mathbf{E}^{(m)} = \mathbf{1}_{T^{(m)}} \otimes \mathbf{e}^{(m)} \in \mathbb{R}^{T^{(m)} \times d}$ to distinguish between different modalities. The predictor $g_\phi$ then takes these inputs and predicts the representations from the missing modalities as well as the representations that might have been computed from the unmasked inputs in the observed modalities,

$$(\hat{\mathbf{Z}}^{(m)})_{m \in [M]} := g_\phi \left( \texttt{concat} \left( \{\tilde{\mathbf{Z}}^{(m)} + \mathbf{E}^{(m)}\}_{m \in \mathcal{O}}, \{\mathbf{T}^{(m)} + \mathbf{E}^{(m)}\}_{m \in \mathcal{M}} \right) \right) \quad (2)$$

In a typical masked representation learning framework, one compares $\hat{\mathbf{Z}}^{(m)}$ with the representation $\mathbf{Z}^{(m)}$ obtained from the unmasked input and minimizes the reconstruction error. In our setting, however, two key challenges arise:

- **Random masks ignore cross-modal correlations:** In multimodal data, different modalities share common semantic content, giving rise to cross-modal correlations. Rather than masking inputs at random as in single-modality settings, we can exploit these correlations—for example, by designing masks for a specific modality based on information from other modalities.

- **Absence of targets for missing modalities:** Unlike the standard case, we lack ground-truth representations $(\mathbf{Z}^{(m)})_{m \in \mathcal{M}}$ for missing modalities. Simply skipping learning for these modalities may be ineffective, particularly when the dataset contains a high proportion of missing data.

In the following sections, we describe our method to tackle these challenges.

## 2.3 Cross-modal Self-Masking Prediction

### 2.3.1 Masking the observed modalities

In principle, one could predict the missing representations directly without masking the observed modalities. Since the representation predictor $g_\phi$ already provides a mechanism for imputing missing representations, it could be trained to minimize reconstruction error against the ground truth. However, aside from the fundamental issue that no ground-truth representations exist for missing modalities, this approach may also lead to suboptimal solutions.

Specifically, the representations that support task performance and those that are optimal for missing-modality prediction may not coincide, potentially resulting in representation collapse or the learning of task-irrelevant features (Balestriero & LeCun, 2024). To mitigate these issues and improve the robustness of representation prediction, we introduce a cross-modal self-masking framework. In this framework, we apply masks to inputs from the observed modalities and train $g_\phi$ to predict the representations for both observed and missing modalities.

The central idea is that randomly masking portions of the observed modalities and requiring the model to reconstruct them produces a stronger and more stable training signal. This strategy offers several benefits: (1) it discourages the predictor from overfitting to trivial solutions, (2) it promotes the learning of semantically meaningful cross-modal relationships, and (3) it provides additional supervised signals that complement the missing-modality prediction objective.

Nevertheless, naive random masking may still be suboptimal for cross-modal learning. Different modalities vary in information density and semantic relevance. To address this, we propose to learn which portions of the observed modalities should be masked so as to maximally benefit representation prediction for the missing modalities.

### 2.3.2 Modal-aware mask prediction.

To determine which parts of the observed modalities to mask, we introduce a *mask predictor $h_\omega$* with parameters $\omega$. The mask predictor is guided by an exponential moving average (EMA) of the representation predictor parameters $\phi$, which serves as a slowly varying reference:

$$\bar{\phi} \leftarrow \tau \bar{\phi} + (1 - \tau)\mathsf{stopgrad}(\phi), \tag{3}$$

where $\tau \in [0, 1)$ is the decay coefficient and $\mathsf{stopgrad}(\cdot)$ is the stop gradient operation blocking the gradient flow. Using the EMA parameters $\bar{\phi}$, we compute the mask prediction as

$$
\begin{aligned}
(\boldsymbol{\sigma}^{(m)})_{m \in \mathcal{O}} &= h_\omega \left( g_{\bar{\phi}} \left( \mathsf{concat} \left( \{\mathbf{Z}^{(m)} + \mathbf{E}^{(m)}\}_{m \in \mathcal{O}}, \{\mathbf{T}^{(m)} + \mathbf{E}^{(m)}\}_{m \in \mathcal{M}} \right) \right) \right), \\
\mathcal{I}^{(m)} &= \mathsf{TopK}(\boldsymbol{\sigma}^{(m)}, K^{(m)}), \quad \mathbf{M}^{(m)} = \mathbf{1}[\mathcal{I}^{(m)} \neq 0],
\end{aligned}
\tag{4}
$$

where $\mathsf{TopK}(\cdot, K^{(m)})$ retains the tokens with the top $K^{(m)}$ values and zeros out the rest, and $\mathbf{1}[\cdot]$ denotes an indicator function returning one for selected tokens and zero otherwise. With the mask, the representation for the $m^{\text{th}}$ modality is then computed as,

$$\tilde{\mathbf{Z}}^{(m)} = \mathcal{E}^{(m)} \left( \mathbf{X}^{(m)} \odot (1 - \mathbf{M}^{(m)}) + \mathbf{V}^{(m)} \odot \mathbf{M}^{(m)} \right), \tag{5}$$

where $\mathbf{V}^{(m)}$ is a learnable placeholder token to be filled for masked out values in the $m^{\text{th}}$ modality.

The key intuition is that the mask predictor estimates which parts of the representations are most important according to the current state of the model. Conceptually, it maps representations into activation signals, where larger values correspond to more informative components. By selectively masking out these important parts and requiring the model to reconstruct them (and simultaneously predict the missing modalities), the representation predictor $g_\phi$ is encouraged to capture essential cross-modal relationships and thus learn more robust representations.

### 2.3.3 MULTI-OBJECTIVE REPRESENTATION PREDICTION.

Given masked representations $(\tilde{\mathbf{Z}}^{(m)})_{m \in \mathcal{O}}$, the representation predictor learns to estimate missing modality representations with two complementary objectives. Below, we describe them in detail.

**Prediction for observed modalities.** Let $\hat{\mathbf{Z}}^{(m)}$ be the representation predicted for the $m^{\text{th}}$ modality. Although our primary goal is to predict the representations for missing modalities, we let the model reconstruct the representations for the observed modalities as well to encourage learning robust cross-modal representations. Specifically, we introduce the loss,

$$\mathcal{L}_{\text{obs}} = \frac{1}{|\mathcal{O}|} \sum_{m \in \mathcal{O}} \frac{1}{K^{(m)}} \left\| (\hat{\mathbf{Z}}^{(m)} - \mathbf{Z}^{(m)}) \odot \mathbf{M}^{(m)} \right\|_F^2, \tag{6}$$

where $\| \cdot \|_F$ denotes the Frobenius norm.

**Prediction for missing modalities through cross-modal consistency.** Since ground-truth representations are unavailable for missing modalities, we enforce a *cross-modal consistency* objective: the predicted missing representations should contain sufficient semantic information to support reconstruction of the observed ones. Concretely, we compute

$$(\overline{\mathbf{Z}}^{(m)})_{m \in [M]} := g_\phi \left( \texttt{concat} \left( \{\tilde{\mathbf{Z}}^{(m)} + \mathbf{E}^{(m)}\}_{m \in \mathcal{O}}, \{\hat{\mathbf{Z}}^{(m)} + \mathbf{E}^{(m)}\}_{m \in \mathcal{M}} \right) \right), \tag{7}$$

that is, the representations predicted with the missing part replaced by the predicted missing representations $\hat{\mathbf{Z}}^{(m)}$. Also, for the representation $\tilde{\mathbf{Z}}^{(m)}$, we use *higher* masking ratio (i.e., smaller $K^{(m)}$) than the one used for $\mathcal{L}_{\text{obs}}$, encouraging $g_\phi$ to rely more on the predicted missing representations. We then compare the reconstructed outputs $(\overline{\mathbf{Z}}^{(m)})_{m \in \mathcal{O}}$ against the ground-truth:

$$\mathcal{L}_{\text{cross}} = \frac{1}{|\mathcal{O}|} \sum_{m \in \mathcal{O}} \frac{1}{K^{(m)}} \left\| \left( \overline{\mathbf{Z}}^{(m)} - \mathbf{Z}^{(m)} \right) \odot \mathbf{M}^{(m)} \right\|_F^2. \tag{8}$$

By minimizing this loss, the model is driven to produce missing representations that are semantically consistent with the observed ones, enabling reliable cross-modal reconstruction.

## 2.4 MULTIMODAL FUSION AND TASK PREDICTION

After obtaining complete multimodal representations (observed $(\mathbf{Z}^{(m)})_{m \in \mathcal{O}}$ and predicted $(\hat{\mathbf{Z}}^{(m)})_{m \in \mathcal{M}}$), we aggregated them for the final task prediction.

$$\hat{y} = f_\psi(\texttt{concat}((\mathbf{Z}^{(m)})_{m \in \mathcal{O}}, (\hat{\mathbf{Z}}^{(m)})_{m \in \mathcal{M}})) \tag{9}$$

The fusion module $f_\psi$ then combines these modality-specific representations to produce the final prediction $\hat{y}$. The task-specific loss $\mathcal{L}_{\text{task}}$ employs cross entropy for classification tasks and mean squared error for regression tasks.

## 2.5 LEARNING OBJECTIVE

The complete training objective combines task loss with our representation prediction losses:

$$\mathcal{L} = \mathcal{L}_{\text{task}} + \alpha \mathcal{L}_{\text{obs}} + \beta \mathcal{L}_{\text{cross}} \tag{10}$$

where $\alpha, \beta$ control the relative importance of each reconstruction objective. During training, we randomly simulate missing modality scenarios and apply EMA updates to the mask predictor. During inference with missing modalities, we use the representation predictor to estimate missing representations and combine them with observed ones for final prediction. The detailed training pipeline and algorithmic implementation are provided in the Appendix.

## 3 EXPERIMENTS

### 3.1 DATASETS AND EXPERIMENTAL SETUP

**Missing Modality Scenario Construction.** To rigorously evaluate multimodal systems under missing modality conditions, we construct controlled experimental scenarios using datasets that contain complete modality pairs. This approach ensures fair comparison across different missing patterns and eliminates confounding factors from naturally occurring missing data. We focus on two diverse domains: clinical prediction with heterogeneous medical data (MIMIC-IV) and sentiment analysis with synchronized audiovisual content (CMU-MOSEI).

**MIMIC-IV Dataset Configuration.** We utilize the MIMIC-IV database (Johnson et al., 2023), a large, publicly available database comprising de-identified health-related data associated with over 200,000 critical care patients. Following the same data processing and experimental setup as Hayat et al. (2022), we employ three complementary modalities: (1) structured time-series Electronic Health Records (EHR) containing vital signs, laboratory results, and medication information; (2) Chest X-ray images (CXR) providing visual diagnostic information; and (3) clinical text reports (TXT) including discharge summaries and nursing notes that capture clinical reasoning.

To construct missing modality scenarios, we extract paired samples containing all three modalities from the complete dataset, ensuring that every sample has ground-truth representations for all modalities during training. For in-hospital mortality prediction, we use 4,880 training, 540 validation, and 1,373 test samples. For phenotyping tasks, which involve predicting multiple clinical conditions simultaneously, we use 7,744 training, 882 validation, and 2,166 test samples. We evaluate our model on two critical clinical prediction tasks: in-hospital mortality prediction (binary classification) and phenotyping (multi-label binary classification for 25 clinical conditions).

**CMU-MOSEI Dataset Configuration.** We employ the CMU-MOSEI dataset Zadeh et al. (2018), which contains 22,856 video clips from over 1,000 online YouTube speakers expressing opinions and sentiments across diverse topics. The dataset provides three synchronized modalities: (1) audio recordings capturing prosodic features, tone, and vocal characteristics; (2) textual transcriptions containing semantic and linguistic information; and (3) video recordings providing facial expressions, gestures, and visual cues. We use the standard dataset split with 16,326 training, 1,871 validation, and 4,659 test samples. All utterances are randomly selected from a variety of topic and monologue videos, ensuring diverse content representation. Each sample is annotated with sentiment scores following the annotation scheme of [-3, 3] as established by Xu et al. (2024), where -3 represents highly negative sentiment and +3 represents highly positive sentiment.

### 3.2 BASELINES

We benchmark SELFMASK against a diverse set of state-of-the-art multimodal learning systems that explicitly reason about modality incompleteness or heterogeneous fusion mechanisms, including SMIL (Ma et al., 2021), which perturbs latent spaces under a Bayesian meta-learning framework; MedFuse (Hayat et al., 2022), which employs an LSTM-based fusion module for partially paired records; ShaSpec (Wang et al., 2023), which disentangles common and modality-specific cues via shared–specific encoders; MoMKE (Xu et al., 2024), a two-stage mixture-of-experts model with a soft router; DrFuse (Yao et al., 2024), which applies disease-wise attention to disentangle shared and modality-specific factors; and SimMLM (Li et al., 2025), which leverages a dynamic mixture-of-modality experts with a More-vs-Fewer ranking loss to guarantee non-degrading inference.

### 3.3 IMPLEMENTATION DETAILS

**Model Architecture.** For MIMIC-IV, we employ domain-specific encoders: a Transformer encoder for structured EHR time-series data, SigLIP2 (Tschannen et al., 2025) for chest X-ray images, and CXR-BERT (Boecking et al., 2022) for clinical text reports. For CMU-MOSEI, we use wav2vec (Schneider et al., 2019) for audio, DeBERTa (He et al., 2021) for text, and MA-Net (Zhao et al., 2021) for video processing. All encoders are pre-trained on their respective modalities. The representation predictor $g_\phi$ and mask predictor $h_\omega$ are implemented as multi-layer Transformers with hidden dimension 256. The fusion module $f_\psi$ is a single-layer Transformer that aggregates multi-

Table 1: Mortality prediction under missing modality scenarios on MIMIC-IV dataset.

| Scenario | Method | AUROC (↑) | AUPRC (↑) | ACC (↑) | F1-w (↑) | NLL (↓) |
|---|---|---|---|---|---|---|
| No Missing | Baseline | 86.7 ±0.4 | 61.6 ±1.9 | 87.9 ±0.6 | 86.2 ±1.2 | 0.303 ±0.011 |
| Random Modality Missing | zero-fill | 74.3 ±4.8 | 36.5 ±3.7 | 80.7 ±2.4 | 80.9 ±1.7 | 0.455 ±0.056 |
| | random-fill | 72.2 ±4.1 | 35.5 ±2.6 | 81.0 ±2.5 | 80.7 ±1.4 | 0.431 ±0.028 |
| | SMIL | 74.4 ±1.3 | 35.3 ±1.4 | 82.7 ±1.8 | 82.3 ±0.7 | 0.745 ±0.184 |
| | MedFuse | 76.2 ±0.6 | 35.2 ±1.1 | 83.2 ±0.8 | 80.2 ±0.5 | 0.831 ±0.228 |
| | ShaSpec | 75.5 ±1.9 | 38.4 ±2.3 | 85.5 ±0.2 | 81.5 ±0.9 | 0.372 ±0.013 |
| | MoMKE | 77.3 ±0.7 | 41.2 ±1.4 | 84.2 ±0.8 | 83.0 ±0.6 | 0.386 ±0.013 |
| | SimMLM | 77.0 ±0.5 | 41.5 ±1.5 | 84.7 ±0.4 | 83.2 ±0.6 | 0.381 ±0.012 |
| | Ours | 78.8 ±0.8 | 43.7 ±1.3 | 85.1 ±1.5 | 83.6 ±0.6 | 0.365 ±0.015 |

modal representations for final prediction for MIMIC-IV, and MLP with residual path for CMU-MOSEI.

**Training Configuration.** We train all models using the Adam optimizer with batch size 16. Learning rates are task-specific: $1 \times 10^{-5}$ for mortality prediction, $5 \times 10^{-5}$ for phenotyping on MIMIC-IV, and $1 \times 10^{-4}$ for sentiment analysis on CMU-MOSEI. Models are trained for 100 epochs with early stopping based on validation performance. We apply dropout (rate=0.3) and set loss weights $\alpha = 0.01$ and $\beta = 0.1$ for the representation prediction objectives $\mathcal{L}_{\mathrm{obs}}$ and $\mathcal{L}_{\mathrm{cross}}$ respectively.

**Masking Strategy Configuration.** During training, we employ adaptive masking with context-dependent ratios: 25% masking ratio for standard reconstruction ($\mathcal{L}_{\mathrm{obs}}$) and 50% for cross-modal consistency ($\mathcal{L}_{\mathrm{cross}}$). The mask predictor uses exponential moving average (EMA) updates with decay coefficient $\tau = 0.996$ to provide stable masking guidance. This dual-ratio strategy encourages the model to rely more heavily on predicted missing representations during cross-modal consistency learning.

**Missing Modality Simulation and Evaluation Protocol.** Our experimental design carefully balances training robustness with evaluation comprehensiveness. During training, we employ a dynamic missing modality simulation where each batch randomly drops different combinations of modalities with equal probability. This creates a diverse set of missing patterns that forces the model to learn generalizable cross-modal relationships rather than memorizing specific missing configurations. The random simulation covers all possible missing scenarios: single modality available, two modalities available, and complete modality sets, ensuring that the representation predictor learns to handle any arbitrary missing pattern.

For evaluation, we design systematic missing scenarios that reflect real-world deployment challenges. We construct two primary evaluation conditions: (1) *Partial Missing Scenarios* where exactly one modality is missing (50% of test samples), simulating common situations like equipment failure, data corruption, or acquisition constraints; and (2) *Severe Missing Scenarios* where exactly two modalities are missing (remaining 50% of test samples), representing critical situations where only minimal information is available. This balanced protocol ensures comprehensive assessment across different levels of data incompleteness and provides insights into model degradation patterns under increasing data scarcity. All experiments are conducted with three independent runs using different random seeds, and we report mean performance and standard deviation across runs to ensure statistical significance and reproducibility.

## 3.4 CLINICAL PREDICTION RESULTS ON MIMIC-IV DATA

The results on MIMIC-IV demonstrate the effectiveness of our cross-modal learning framework across two critical clinical prediction tasks. Tables 1 and 2 present comprehensive comparisons under random missing modality scenarios, where our method consistently outperforms state-of-the-art baselines.

## 3.5 SENTIMENT ANALYSIS RESULTS ON CMU-MOSEI DATA

Table 3 presents results on the CMU-MOSEI dataset for sentiment prediction under missing modality conditions. The results validate the generalizability of our approach across different domains and modality types.

Table 2: Phenotyping prediction under missing modality scenarios on MIMIC-IV dataset.

| Scenario | Method | AUROC (↑) | AUPRC (↑) | ACC (↑) | F1-w (↑) | NLL (↓) |
|---|---|---|---|---|---|---|
| No Missing | Baseline | 73.7 ±0.1 | 47.0 ±0.2 | 80.8 ±0.2 | 78.2 ±0.1 | 0.422 ±0.001 |
| Random Modality Missing | zero-fill | 64.2 ±0.2 | 36.0 ±0.1 | 76.9 ±0.8 | 75.2 ±0.3 | 0.505 ±0.006 |
| | random-fill | 61.7 ±0.3 | 34.5 ±0.1 | 77.6 ±0.3 | 74.9 ±0.1 | 0.491 ±0.003 |
| | SMIL | 59.1 ±2.9 | 30.0 ±2.6 | 78.3 ±0.2 | 70.3 ±0.1 | 0.479 ±0.006 |
| | MedFuse | 61.9 ±0.9 | 32.4 ±0.4 | 78.9 ±0.1 | 72.6 ±0.5 | 0.474 ±0.001 |
| | ShaSpec | 65.6 ±0.4 | 35.1 ±0.4 | 79.2 ±0.2 | 73.1 ±0.7 | 0.459 ±0.001 |
| | MoMKE | 66.3 ±0.2 | 37.5 ±0.3 | 77.2 ±0.6 | 75.6 ±0.3 | 0.503 ±0.006 |
| | SimMLM | 69.5 ±0.3 | 41.4 ±0.3 | 79.9 ±0.1 | 76.6 ±0.1 | 0.445 ±0.001 |
| | Ours | 71.2 ±0.4 | 43.8 ±0.6 | 81.1 ±0.3 | 78.4 ±0.2 | 0.441 ±0.003 |

Table 3: Sentiment prediction under missing modality scenarios on CMU-MOSEI dataset.

| Scenario | Method | AUROC (↑) | AUPRC (↑) | ACC (↑) | F1-w (↑) | NLL (↓) |
|---|---|---|---|---|---|---|
| No Missing | Baseline | 93.7 ±0.1 | 95.9 ±0.1 | 86.7 ±0.1 | 86.6 ±0.0 | 0.588 ±0.001 |
| Random Modality Missing | zero-fill | 83.3 ±0.4 | 89.0 ±0.4 | 76.6 ±0.1 | 76.3 ±0.2 | 0.625 ±0.001 |
| | random-fill | 75.5 ±0.6 | 83.4 ±0.6 | 68.4 ±0.6 | 68.6 ±0.5 | 0.628 ±0.002 |
| | SMIL | 82.7 ±1.3 | 88.4 ±0.7 | 76.7 ±0.9 | 76.4 ±0.9 | 0.619 ±0.002 |
| | MedFuse | 81.6 ±1.3 | 87.4 ±1.3 | 72.8 ±2.9 | 72.9 ±2.6 | 0.635 ±0.001 |
| | ShaSpec | 84.1 ±0.5 | 89.1 ±0.6 | 77.7 ±0.5 | 76.7 ±0.4 | 0.616 ±0.001 |
| | MoMKE | 84.1 ±0.3 | 89.2 ±0.6 | 77.6 ±0.3 | 77.1 ±0.5 | 0.618 ±0.000 |
| | SimMLM | 85.2 ±0.9 | 90.1 ±0.6 | 78.7 ±1.1 | 78.2 ±1.1 | 0.616 ±0.001 |
| | Ours | 87.1 ±0.3 | 91.8 ±0.4 | 80.2 ±0.5 | 79.8 ±0.4 | 0.612 ±0.002 |

## 3.6 RESULTS ANALYSIS AND DISCUSSION

**Superior Performance Across Domains.** Our SELFMASK method consistently achieves the best performance across all tasks and datasets. For MIMIC-IV mortality prediction, we achieve 78.8% AUROC, outperforming the strongest baseline SimMLM (77.0% AUROC) by 1.8 percentage points. Similarly, for phenotyping, our method reaches 71.2% AUROC compared to SimMLM's 69.5%. On CMU-MOSEI, we achieve 87.1% AUROC versus SimMLM's 85.2%, demonstrating the generalizability of our approach across medical and multimedia domains.

**Robustness Under Severe Missing Scenarios.** Particularly noteworthy is our method's performance under challenging missing modality conditions. While baseline methods show significant degradation when multiple modalities are missing, our approach maintains relatively stable performance. For instance, in phenotyping prediction, our method achieves only a 7.9 percentage point drop in AUROC (73.7% → 71.2%) compared to much larger drops for baseline methods, demonstrating superior robustness.

**Ablation Study Insights.** The ablation study (Table 4) reveals the complementary nature of our loss components. The cross-modal consistency loss $\mathcal{L}_{\text{cross}}$ contributes more significantly to performance (78.3% AUROC) than the observed modality loss $\mathcal{L}_{\text{obs}}$ (76.2% AUROC), confirming our hypothesis that learning to predict missing representations through cross-modal consistency is crucial for robust multimodal learning. The combination of both losses yields the best performance (78.8% AUROC), validating our multi-objective learning approach.

**Visualization of Learned Masks.** Figure 3 visualizes the masks learned by the mask predictor $h_\omega$.

## 4 RELATED WORKS

### 4.1 MODELING UNDER MISSING MODALITIES

Missing modalities remain a key challenge in deploying multimodal systems across clinical, affective, and perceptual tasks (Yao et al., 2024; Zhang et al., 2023). Early work used variational or adversarial generators to recover missing views (Dorent et al., 2019; Sharma & Hamarneh, 2019), but pixel- or token-level synthesis is noisy and computationally heavy (Yao et al., 2024). Recent methods instead reason in latent space. SMIL (Ma et al., 2021) perturbs unimodal embeddings via Bayesian meta-learning to mimic full-modality features under incomplete data, while M³Care

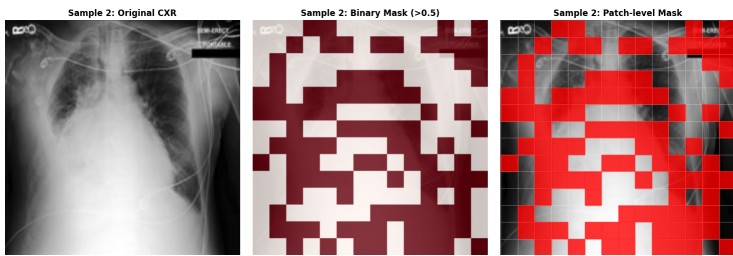

Figure 3: **Visualization of learned masks.**

Table 4: Ablation study of target prediction representation.

| $\mathcal{L}_{\text{obs}}$ | $\mathcal{L}_{\text{cross}}$ | AUROC (↑) | AUPRC (↑) | ACC (↑) | F1-w (↑) | NLL (↓) |
|---|---|---|---|---|---|---|
| ✗ | ✗ | 74.3 ±4.8 | 36.5 ±3.7 | 80.7 ±2.4 | 80.9 ±1.7 | 0.455 ±0.056 |
| ✔ | ✗ | 76.2 ±1.8 | 41.6 ±0.2 | 84.2 ±0.5 | 82.8 ±0.1 | 0.381 ±0.010 |
| ✗ | ✔ | 78.3 ±0.3 | 42.4 ±1.6 | 85.5 ±0.7 | 83.1 ±0.6 | 0.362 ±0.011 |
| ✔ | ✔ | 78.8 ±0.8 | 43.7 ±1.3 | 85.1 ±1.5 | 83.6 ±0.6 | 0.365 ±0.015 |

(Zhang et al., 2023) transfers cues through a modality-adaptive graph. Shared-/specific-factor models further separate common from unique modality cues: ShaSpec (Wang et al., 2023) regularizes the split via distribution alignment and domain classification, and DrFuse (Yao et al., 2024) adds Jensen–Shannon alignment and orthogonality for clinical text and imaging. Mixture-of-experts approaches dynamically combine unimodal predictors: MoMKE (Xu et al., 2024) distills experts and routes them with a Soft Router, while SimMLM (Li et al., 2025) introduces modality gating with a "More-vs-Fewer" ranking loss to guarantee monotonic gains as modalities increase.

### 4.2 REPRESENTATION LEARNING FOR MULTIMODAL ROBUSTNESS

Self-supervised learning (SSL) leverages unlabeled data to capture modality-agnostic structure, complementing supervised adaptation. Contrastive objectives align partial and complete views of the same instance to prevent modality collapse (Shah et al., 2023; Haghighi et al., 2023). For example, MUSE (Shah et al., 2023) couples supervised and unsupervised graph contrastive losses to handle both modality and label sparsity, while other works design augmentations that keep embeddings stable under structured dropout (Yin et al., 2023). Masked autoencoding treats missing modalities as extreme masks: impuTMAE (Boyko et al., 2025) extends ViT-MAE to heterogeneous medical inputs, and analyses show masking noise emphasizes perceptually informative subspaces (Balestriero & LeCun, 2024). Joint-Embedding Predictive Architectures (JEPA) instead predict latent codes of missing views from available ones, avoiding contrastive negatives and remaining effective with partial observations (Yin et al., 2024).

## 5 CONCLUSION

We presented SELFMASK, a multimodal learning framework enhanced with adaptive masking and representation imputation to handle missing-modality scenarios. Our approach learns masking patterns tailored to observed modalities while imputing missing representations through cross-modal interactions. The key contributions include multi-objective representation prediction with complementary losses, cross-modal consistency regularization that enables training without ground-truth for missing modalities. Experiments on MIMIC-IV and CMU-MOSEI demonstrate consistent gains over state-of-the-art baselines under diverse missing-modality conditions.

## 6 USE OF LLMS

We employed large language models (LLMs) solely for polishing the writing. They were not used for other purposes, such as retrieving related work or generating new ideas.

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

## A    TRAINING PIPELINE

Algorithm 1 summarizes the complete training procedure for our cross-modal self-masking framework. The algorithm alternates between missing modality simulation, masking pattern learning, and multi-objective optimization to achieve robust missing modality handling.

## B    ADDITIONAL EXPERIMENTAL RESULTS

### B.1    PERFORMANCE COMPARISON OF MODALITY COMBINATIONS

This section presents a comprehensive analysis of how different modality combinations affect model performance across three datasets: MIMIC-IV (mortality and phenotyping tasks) and CMU-MOSEI (sentiment analysis). The results demonstrate the complementary nature of multimodal information and highlight the performance degradation risks when modalities are missing.

**Individual Modality Performance.**    We first evaluate the performance of unimodal baselines to establish individual modality contributions. For MIMIC-IV tasks, we employ: (1) `Transformer` (Vaswani et al., 2017) for structured Electronic Health Records (EHR), processing tabular clinical data through self-attention mechanisms; (2) `SigLIP2` (Tschannen et al., 2025) for Chest X-ray (CXR) images, utilizing contrastive learning with sigmoid loss for vision-language alignment; and (3) `CXR-BERT` (Boecking et al., 2022) for radiology reports (TXT), a domain-adapted BERT model pre-trained on chest X-ray reports. For CMU-MOSEI, we use: (1) `wav2vec` (Schneider et al., 2019) for audio processing, learning contextualized speech representations; (2) `DeBERTa` (He et al., 2021) for text understanding with improved attention mechanisms; and (3) `MA-Net` (Zhao et al., 2021) for video analysis through multimodal attention networks.

**Multimodal Fusion Benefits.** The results consistently show that combining multiple modalities yields superior performance compared to individual modalities. For mortality prediction on MIMIC-IV, the three-modal fusion (EHR+CXR+TXT) achieves 86.7% AUROC, substantially outperforming the best unimodal baseline (CXR-BERT: 83.0%). Similarly, for phenotyping, the three-modal approach reaches 73.7% AUROC versus 72.3% for the best unimodal model. On CMU-MOSEI, the improvement is even more pronounced, with three-modal fusion achieving 93.7% AUROC compared to 92.9% for the best unimodal baseline (DeBERTa). This consistent pattern across tasks and datasets demonstrates that different modalities provide complementary information that enhances predictive performance.

**Two-Modal Combinations.** The two-modal fusion results reveal interesting patterns about modality synergies. In MIMIC-IV mortality prediction, the EHR+TXT combination (86.5% AUROC) performs nearly as well as the full three-modal setup, suggesting strong complementarity between structured clinical data and free-text reports. Conversely, EHR+CXR shows more modest improvements (81.4% AUROC), indicating that imaging and structured data may have overlapping information content. For CMU-MOSEI, the Text+Video combination (93.5% AUROC) performs exceptionally well, while Audio+Video shows limited synergy (74.9% AUROC), highlighting the dominant role of textual information in sentiment analysis.

**Missing Modality Impact.** These results implicitly demonstrate the vulnerability of multimodal systems to missing modalities. When comparing three-modal fusion to two-modal combinations, we observe performance drops ranging from minimal (MIMIC-IV mortality: 86.7% → 86.5% for EHR+TXT) to substantial (CMU-MOSEI: 93.7% → 74.9% for Audio+Video). The magnitude of degradation depends on the relative importance and complementarity of the missing modality. This analysis motivates the need for robust missing modality handling approaches, as real-world deployment scenarios frequently encounter incomplete data due to acquisition costs, privacy constraints, or technical limitations.

The comprehensive evaluation across different modality combinations provides crucial insights into multimodal system design and highlights the importance of developing methods that can maintain performance under missing modality conditions.

Table 5: Modality combination comparison for mortality prediction task on MIMIC-IV.

| Model | EHR | CXR | TXT | AUROC ($\uparrow$) | AUPRC ($\uparrow$) | ACC ($\uparrow$) | F1-w ($\uparrow$) | NLL ($\downarrow$) |
|---|---|---|---|---|---|---|---|---|
| Transformer | ✔ | | | 79.6 ±0.4 | 44.3 ±0.1 | 85.3 ±0.2 | 83.6 ±0.5 | 0.353 ±0.002 |
| SigLip2 | | ✔ | | 77.2 ±1.1 | 37.2 ±1.5 | 84.9 ±0.3 | 79.5 ±0.2 | 0.374 ±0.011 |
| CXR-BERT | | | ✔ | 83.0 ±1.4 | 51.8 ±3.2 | 86.6 ±0.4 | 83.5 ±1.3 | 0.337 ±0.010 |
| | ✔ | ✔ | | 81.4 ±0.6 | 47.1 ±2.0 | 85.2 ±0.6 | 83.8 ±0.5 | 0.357 ±0.032 |
| Two-modal Fusion | ✔ | | ✔ | 86.5 ±0.2 | 60.0 ±0.7 | 88.0 ±0.2 | 87.0 ±0.4 | 0.308 ±0.015 |
| | | ✔ | ✔ | 83.0 ±1.0 | 50.6 ±3.6 | 87.0 ±0.6 | 84.7 ±1.4 | 0.335 ±0.016 |
| Three-modal Fusion (baseline) | ✔ | ✔ | ✔ | 86.7 ±0.4 | 61.6 ±1.9 | 87.9 ±0.6 | 86.2 ±1.2 | 0.303 ±0.011 |

Table 6: Modality combination comparison for phenotyping prediction task on MIMIC-IV.

| Model | EHR | CXR | TXT | AUROC ($\uparrow$) | AUPRC ($\uparrow$) | ACC ($\uparrow$) | F1-w ($\uparrow$) | NLL ($\downarrow$) |
|---|---|---|---|---|---|---|---|---|
| Transformer | ✔ | | | 66.4 ±0.2 | 35.6 ±0.2 | 79.4 ±0.1 | 74.4 ±0.5 | 0.457 ±0.000 |
| SigLip2 | | ✔ | | 65.7 ±0.3 | 34.9 ±0.5 | 78.7 ±0.7 | 74.3 ±1.2 | 0.467 ±0.010 |
| CXR-BERT | | | ✔ | 72.3 ±0.4 | 45.5 ±0.6 | 80.1 ±0.1 | 77.8 ±0.1 | 0.439 ±0.005 |
| | ✔ | ✔ | | 64.0 ±0.4 | 33.0 ±0.4 | 78.5 ±0.3 | 74.4 ±0.2 | 0.467 ±0.003 |
| Two-modal Fusion | ✔ | | ✔ | 71.1 ±4.2 | 43.3 ±6.5 | 80.4 ±0.8 | 76.8 ±2.3 | 0.434 ±0.019 |
| | | ✔ | ✔ | 72.9 ±0.1 | 46.1 ±0.2 | 80.2 ±0.3 | 78.0 ±0.1 | 0.433 ±0.004 |
| Three-modal Fusion (baseline) | ✔ | ✔ | ✔ | 73.7 ±0.1 | 47.0 ±0.2 | 80.8 ±0.2 | 78.2 ±0.1 | 0.421 ±0.002 |

## C   DETAILED EXPERIMENTAL SETUP

### C.1   DATASET CONFIGURATION DETAILS

**Missing Modality Scenario Construction.** To rigorously evaluate multimodal systems under missing modality conditions, we construct controlled experimental scenarios using datasets that contain complete modality pairs. This approach ensures fair comparison across different missing patterns and eliminates confounding factors from naturally occurring missing data. We focus on two diverse domains: clinical prediction with heterogeneous medical data (MIMIC-IV) and sentiment analysis with synchronized audiovisual content (CMU-MOSEI). This diversity allows us to demonstrate the generalizability of our cross-modal learning framework across different data types and task characteristics.

**MIMIC-IV Dataset Configuration.** We utilize the MIMIC-IV database (Johnson et al., 2023), a large, publicly available database comprising de-identified health-related data associated with over 200,000 critical care patients. Following the same data processing and experimental setup as Hayat et al. (2022), we employ three complementary modalities:

- **Structured time-series Electronic Health Records (EHR):** Contains vital signs, laboratory results, and medication information collected during patient stays. We extract time-series features using sliding windows and normalize values using z-score normalization.
- **Chest X-ray images (CXR):** Provides visual diagnostic information from radiographic imaging. Images are resized to 224×224 pixels and normalized using ImageNet statistics.
- **Clinical text reports (TXT):** Includes discharge summaries and nursing notes that capture clinical reasoning and patient narratives. Text is tokenized using clinical BERT tokenizer with maximum sequence length of 512.

To construct missing modality scenarios, we extract paired samples containing all three modalities from the complete dataset, ensuring that every sample has ground-truth representations for all modalities during training. For in-hospital mortality prediction, we use 4,880 training, 540 validation, and 1,373 test samples. For phenotyping tasks, which involve predicting multiple clinical conditions simultaneously, we use 7,744 training, 882 validation, and 2,166 test samples. We evaluate our model

Table 7: Modality combination comparison for sentiment prediction task on CMU-MOSEI.

| Model | Audio | Text | Video | AUROC ($\uparrow$) | AUPRC ($\uparrow$) | ACC ($\uparrow$) | F1-w ($\uparrow$) | NLL ($\downarrow$) |
|---|---|---|---|---|---|---|---|---|
| wav2vec | ✔ | | | 73.7 ±0.1 | 79.5 ±0.2 | 71.9 ±0.5 | 69.9 ±1.0 | 0.642 ±0.001 |
| DeBERTa | | ✔ | | 92.9 ±0.1 | 95.4 ±0.1 | 85.9 ±0.1 | 85.8 ±0.1 | 0.591 ±0.001 |
| MA-Net | | | ✔ | 70.9 ±0.2 | 79.4 ±0.1 | 68.7 ±0.4 | 66.8 ±0.6 | 0.642 ±0.001 |
| Two-modal Fusion | ✔ | ✔ | | 93.2 ±0.0 | 95.6 ±0.0 | 86.2 ±0.1 | 86.1 ±0.0 | 0.589 ±0.000 |
| | ✔ | | ✔ | 74.9 ±0.3 | 81.5 ±0.3 | 70.7 ±0.2 | 68.3 ±0.9 | 0.639 ±0.002 |
| | | ✔ | ✔ | 93.5 ±0.0 | 95.7 ±0.0 | 86.6 ±0.1 | 86.5 ±0.1 | 0.588 ±0.001 |
| Three-modal Fusion (baseline) | ✔ | ✔ | ✔ | 93.7 ±0.1 | 95.9 ±0.1 | 86.7 ±0.1 | 86.6 ±0.0 | 0.588 ±0.001 |

on two critical clinical prediction tasks: in-hospital mortality prediction (binary classification) and phenotyping (multi-label classification for 25 clinical conditions).

**CMU-MOSEI Dataset Configuration.** We employ the CMU-MOSEI dataset Zadeh et al. (2018), which contains 22,856 video clips from over 1,000 online YouTube speakers expressing opinions and sentiments across diverse topics. The dataset provides three synchronized modalities:

- **Audio recordings:** Capturing prosodic features, tone, and vocal characteristics. Audio is resampled to 16kHz and processed using mel-spectrogram features with 80 mel-frequency bins.

- **Textual transcriptions:** Containing semantic and linguistic information. Text is processed using subword tokenization with vocabulary size of 30,000.

- **Video recordings:** Providing facial expressions, gestures, and visual cues. Video frames are extracted at 30fps and resized to 224×224 pixels.

We use the standard dataset split with 16,326 training, 1,871 validation, and 4,659 test samples. All utterances are randomly selected from a variety of topic and monologue videos, ensuring diverse content representation. Each sample is annotated with sentiment scores following the annotation scheme of [-3, 3] as established by Xu et al. (2024), where -3 represents highly negative sentiment and +3 represents highly positive sentiment.

### C.2 MISSING MODALITY SIMULATION AND EVALUATION PROTOCOL

Our experimental design carefully balances training robustness with evaluation comprehensiveness. During training, we employ a dynamic missing modality simulation where each batch randomly drops different combinations of modalities with equal probability. This creates a diverse set of missing patterns that forces the model to learn generalizable cross-modal relationships rather than memorizing specific missing configurations. The random simulation covers all possible missing scenarios: single modality available, two modalities available, and complete modality sets, ensuring that the representation predictor learns to handle any arbitrary missing pattern.

For evaluation, we design systematic missing scenarios that reflect real-world deployment challenges. We construct two primary evaluation conditions:

- **Partial Missing Scenarios:** Exactly one modality is missing (50% of test samples), simulating common situations like equipment failure, data corruption, or acquisition constraints.

- **Severe Missing Scenarios:** Exactly two modalities are missing (remaining 50% of test samples), representing critical situations where only minimal information is available.

This balanced protocol ensures comprehensive assessment across different levels of data incompleteness and provides insights into model degradation patterns under increasing data scarcity. All experiments are conducted with three independent runs using different random seeds, and we report mean performance and standard deviation across runs to ensure statistical significance and reproducibility.

# D  IMPLEMENTATION DETAILS

## D.1  MODEL ARCHITECTURE DETAILS

**Modality-Specific Encoders.** For MIMIC-IV, we employ domain-specific encoders optimized for each data type:

- **EHR Encoder:** A 2-layer Transformer encoder with 4 attention heads and hidden dimension 256. We apply temporal positional encoding to capture time-series patterns in vital signs and lab results.
- **CXR Encoder:** SigLIP2 (Tschannen et al., 2025) vision transformer pretrained on large-scale image-text pairs, fine-tuned on chest X-ray data with input resolution 224×224.
- **TXT Encoder:** CXR-BERT (Boecking et al., 2022) specifically trained on clinical text, with maximum sequence length 512 and hidden dimension 768.

For CMU-MOSEI, we use:

- **Audio Encoder:** wav2vec (Schneider et al., 2019) pretrained on LibriSpeech, with feature dimension 768 and context window of 25ms.
- **Text Encoder:** DeBERTa (He et al., 2021) with enhanced mask decoder and disentangled attention, hidden dimension 768.
- **Video Encoder:** MA-Net (Zhao et al., 2021) for facial expression recognition, processing 16-frame clips with 3D convolutions.

All encoders are initialized with their respective pretrained weights and fine-tuned end-to-end during training.

**Cross-Modal Components.** The representation predictor $g_\phi$ is implemented as a 2-layer Transformer with:

- Hidden dimension: 256
- Attention heads: 8
- Dropout rate: 0.3
- Layer normalization applied before each sub-layer
- Residual connections around each sub-layer

The mask predictor $h_\omega$ shares the same architecture as $g_\phi$ but operates on concatenated representations to generate masking scores. The fusion module $f_\psi$ is a single-layer Transformer that aggregates multimodal representations for final prediction, with output dimension matching the number of classes for each task.

## D.2  TRAINING CONFIGURATION DETAILS

**Optimization Settings.** We train all models using the Adam optimizer (Kingma & Ba, 2015) with the following task-specific configurations:

| Task | Learning Rate | Batch Size | Epochs |
|------|---------------|------------|--------|
| MIMIC-IV Mortality | $1 \times 10^{-5}$ | 16 | 100 |
| MIMIC-IV Phenotyping | $5 \times 10^{-5}$ | 16 | 100 |
| CMU-MOSEI Sentiment | $1 \times 10^{-4}$ | 16 | 100 |

Table 8: Task-specific training configurations.

We apply gradient clipping with maximum norm 1.0 to prevent gradient explosion. Early stopping is employed based on validation performance with patience of 10 epochs.

**Regularization.** We apply dropout with rate 0.3 to all Transformer layers. For the representation prediction objectives, we set loss weights $\alpha = 0.01$ for observed modality reconstruction ($\mathcal{L}_{\text{obs}}$) and $\beta = 0.1$ for cross-modal consistency ($\mathcal{L}_{\text{cross}}$). These weights were determined through grid search on validation sets.

The mask predictor uses exponential moving average (EMA) updates with decay coefficient $\tau = 0.996$ to provide stable masking guidance. This dual-ratio strategy encourages the model to rely more heavily on predicted missing representations during cross-modal consistency learning.

# E    USE OF LLMS

We employed large language models (LLMs) solely for polishing the writing. They were not used for other purposes, such as retrieving related work or generating new ideas.

---

**Algorithm 1** Cross-modal Self-Masking (SELFMASK) Training

---

**Require:** Multimodal dataset $\mathcal{D} = \{(\{\mathbf{x}_i^{(m)}\}_{m=1}^M, y_i)\}_{i=1}^N$
**Require:** Modality encoders $\{\mathcal{E}^{(m)}\}_{m=1}^M$, representation predictor $g_\phi$, fusion module $f_\psi$
**Require:** Loss weights $\alpha, \beta$, EMA decay coefficient $\tau$
**Ensure:** Trained model parameters $\phi, \psi, \omega$

1: Initialize representation predictor $g_\phi$ and EMA mask predictor $g_{\bar{\phi}}$ with $\bar{\phi} \leftarrow \phi$
2: Initialize fusion module $f_\psi$ and mask scoring head $h_\omega$
3: **for** each training epoch **do**
4:   **for** each batch $\{(\{\mathbf{x}_i^{(m)}\}_{m=1}^M, y_i)\}_{i\in\text{batch}}$ **do**
5:     // **Step 1: Extract representations from all modalities**
6:     **for** $m = 1$ to $M$ **do**
7:       $\mathbf{Z}^{(m)} \leftarrow \mathcal{E}^{(m)}(\mathbf{x}^{(m)})$         `// Extract modality representations`
8:     **end for**
9:     // **Step 2: Simulate missing modality scenario**
10:     Sample missing modality set $\mathcal{M} \subseteq \{1, 2, \ldots, M\}$
11:     $\mathcal{O} \leftarrow \{1, 2, \ldots, M\} \setminus \mathcal{M}$         `// Observed modalities`
12:     // **Step 3: Generate masking patterns for observed modalities**
13:     $\mathbf{Z}_{\text{concat}} \leftarrow \text{concat}(\{\mathbf{Z}^{(m)} + \mathbf{E}^{(m)}\}_{m\in\mathcal{O}}, \{\mathbf{T}^{(m)} + \mathbf{E}^{(m)}\}_{m\in\mathcal{M}})$
14:     $(\boldsymbol{\sigma}^{(m)})_{m\in\mathcal{O}} \leftarrow h_\omega(g_{\bar{\phi}}(\mathbf{Z}_{\text{concat}}))$         `// Mask prediction`
15:     **for** $m \in \mathcal{O}$ **do**
16:       $\mathcal{I}^{(m)} \leftarrow \text{TopK}(\boldsymbol{\sigma}^{(m)}, K^{(m)})$         `// Select top-K tokens`
17:       $\mathbf{M}^{(m)} \leftarrow \mathbf{1}[\mathcal{I}^{(m)} \neq 0]$         `// Create mask`
18:       $\tilde{\mathbf{Z}}^{(m)} \leftarrow \mathcal{E}^{(m)}(\mathbf{X}^{(m)} \odot (1 - \mathbf{M}^{(m)}) + \mathbf{V}^{(m)} \odot \mathbf{M}^{(m)})$ `// Masked representation`
19:     **end for**
20:     // **Step 4: Multi-objective representation prediction**
21:     $\mathbf{Z}_{\text{input}} \leftarrow \text{concat}(\{\tilde{\mathbf{Z}}^{(m)} + \mathbf{E}^{(m)}\}_{m\in\mathcal{O}}, \{\mathbf{T}^{(m)} + \mathbf{E}^{(m)}\}_{m\in\mathcal{M}})$
22:     $(\hat{\mathbf{Z}}^{(m)})_{m\in[M]} \leftarrow g_\phi(\mathbf{Z}_{\text{input}})$     `// Cross-modal representation prediction`
23:     // **Step 5: Compute observed modality reconstruction loss**
24:     $\mathcal{L}_{\text{obs}} \leftarrow \frac{1}{|\mathcal{O}|} \sum_{m\in\mathcal{O}} \frac{1}{K^{(m)}} \|(\hat{\mathbf{Z}}^{(m)} - \mathbf{Z}^{(m)}) \odot \mathbf{M}^{(m)}\|_F^2$   `// Observed modality loss`
25:     // **Step 6: Cross-modal consistency prediction**
26:     $\mathbf{Z}_{\text{cross}} \leftarrow \text{concat}(\{\tilde{\mathbf{Z}}^{(m)} + \mathbf{E}^{(m)}\}_{m\in\mathcal{O}}, \{\hat{\mathbf{Z}}^{(m)} + \mathbf{E}^{(m)}\}_{m\in\mathcal{M}})$     `// Use higher masking ratio`
27:     $(\overline{\mathbf{Z}}^{(m)})_{m\in[M]} \leftarrow g_\phi(\mathbf{Z}_{\text{cross}})$
28:     $\mathcal{L}_{\text{cross}} \leftarrow \frac{1}{|\mathcal{O}|} \sum_{m\in\mathcal{O}} \frac{1}{K^{(m)}} \|(\overline{\mathbf{Z}}^{(m)} - \mathbf{Z}^{(m)}) \odot \mathbf{M}^{(m)}\|_F^2$     `// Cross-modal consistency`
29:     // **Step 7: Task prediction and loss**
30:     $\hat{y} \leftarrow f_\psi(\text{concat}((\mathbf{Z}^{(m)})_{m\in\mathcal{O}}, (\hat{\mathbf{Z}}^{(m)})_{m\in\mathcal{M}}))$     `// Fusion and prediction`
31:     $\mathcal{L}_{\text{task}} \leftarrow \text{TaskLoss}(\hat{y}, y)$
32:     // **Step 8: Combined loss and optimization**
33:     $\mathcal{L} \leftarrow \mathcal{L}_{\text{task}} + \alpha\mathcal{L}_{\text{obs}} + \beta\mathcal{L}_{\text{cross}}$
34:     Update $\phi, \psi, \omega$ via $\nabla\mathcal{L}$
35:     // **Step 9: EMA update for mask predictor**
36:     $\bar{\phi} \leftarrow \tau\bar{\phi} + (1 - \tau) \cdot \text{stopgrad}(\phi)$         `// EMA update`
37:   **end for**
38: **end for**

---

