# OpenReview forum: "SelfMask: Cross-modal Self-Masking for Multimodal Representation Learning in Missing Modality Scenarios"
_ICLR.cc/2026/Conference — ICLR 2026 Conference Desk Rejected Submission_

### Official Review · Reviewer_pQCE · 2025-10-31

**Soundness:** 3
**Presentation:** 3
**Contribution:** 2
**Rating:** 2
**Confidence:** 3

**Summary:**

This paper introduces SELFMASK, a framework for learning robust multimodal representations when modalities are missing during deployment. The goal is to move beyond random masking by learning adaptive masking patterns that are informed by cross-modal relationships, while imputing missing modality representations through a cross-modal consistency objective. The approach consists of three main components:​

- Adaptive Mask Prediction: A mask predictor learns which parts of observed modalities to mask based on an exponential moving average (EMA) of the representation predictor​ weights and leverages the reconstruction of the learned mask in the loss (L_obs)
- Cross-Modal Representation Imputation: Missing modality representations are predicted from observed modalities and trained via reconstruction of the observed modality representations​ from the predicted via a cross-modal consistency loss (L_cross)
- Multimodal Fusion and Task Prediction: The predicted and observed modalities are agregated to predict the final result (L_task)

**Strengths:**

1. Excellent Presentation and Figures: The paper is well written, easy to understand, and follow, and the Figures and equations are informative and convey the ideas of the paper well.

2.  Rigorous Experimentation: Experiments were averaged over independent runs with standard deviations reported. The tables are clear to understand, and the model shows consistent improvements over baselines.

3. Elegant Method: The model combines various ideas well, including masked representation reconstruction for intramodal and cross-modal feature learning, as well as missing modality feature imputation.

**Weaknesses:**

1. Limited Novelty: The core components of SelfMask closely resemble existing approaches in missing modality learning. Specifically:
   - The representation-level imputation approach is similar to ActionMAE’s training framework, CMAT's cross-modal translation framework
   - Masked autoencoder approaches for multimodal learning (M3AE, MultiMAE) already explore similar masking strategies. There is not enough evidence or support to claim learned masking over random masking is a significant contribution (e.g, ablations, motivating examples, visualizations, etc.)
   - The combination of learning each modality's unique features as well as imputing missing ones through a reconstructed feature is also present in Robult
The authors should clearly differentiate their contributions from these existing works and provide a comparative analysis to justify the claimed novelty.

CMAT: Park et al  "Cross-modal alignment and translation for missing modality action recognition" (ACM Computer Vision and Image Understanding, 2023)

ActionMAE: Park et al. “Towards Good Practices for Missing Modality Robust Action Recognition” (AAAI 2023)

M3AE: Geng et al. "Multimodal Masked Autoencoders Learn Transferable Representations" (2022)

MultiMAE: Bachmann et al MultiMAE: Multi-modal Multi-task Masked Autoencoders (ECCV 2022)

Robult: Nguyen et al: Robult: Leveraging Redundancy and Modality Specific Features for Robust Multimodal Learning (IJCAI 2025)

2.  Limited Theoretical Analysis: The paper lacks a theoretical justification for why the proposed EMA-based mask predictor should learn informative masks​. The connection between mask informativeness and cross-modal consistency is relatively intuitive but not rigorously established

3.  Hyperparameter Sensitivity and Complexity: The framework introduces multiple hyperparameters (α=0.01, β=0.1, τ=0.996, different masking ratios) without comprehensive sensitivity analysis​ or intuitive justification for them. The dual masking ratios (25% for Lobs, 50% for Lcross) appear somewhat arbitrary and may require task-specific tuning​.

4. Computational Overhead Analysis: Some discussion of the comptuational overhead of this method would be beneficial. The EMA mask predictor and dual forward passes increase computational cost during training, but this overhead is not quantified​. No analysis of inference time complexity when handling missing modalities in deployment

5. Evaluations and Visualizations: The analysis of performance across different missing-modality scenarios is limited. In particular, it’s unclear how the model behaves when certain modality combinations are consistently unavailable or when the proportion of missing data varies. Figure 3 lacks clarity—why did the model choose to mask those specific parts? Were those regions highly informative during training (e.g., indicative of disease) or relatively uninformative? The paper would benefit from clearer and more interpretable visualizations, as well as a systematic failure mode analysis. For example, which test instances did the model handle well despite missing modalities where previous approaches failed? How did the self-mask design contribute to this success? Conversely, on what types of instances does the model still struggle?

**Questions:**

*Clarification Questions:*

1. Are the learned missing tokens (the pink section in Figure 2) shared across all modalities, or are they learned separately for each modality? Additionally, is V^m used to represent missing modalities, or is it only employed to fill in the masks for observed modalities that were intentionally masked during training?

2. Is the sequence length fixed for each modality and input sample, or does it vary depending on the sample or modality? For instance, would a 10-second video and a 30-second video both be represented by the same number of tokens, or would their token counts differ? How is this handled in the model across different samples and modalities?

*Thought Experiments/Extensions:*

3. Mask Informativeness: How do you ensure that the learned masks are indeed more informative than random masks? Can you provide a quantitative or qualitative analysis of mask quality beyond downstream performance, such as an ablation with random masks instead of learned masks.
4. Cross-Modal Dependence: How does the method perform when modalities have minimal shared information or when cross-modal correlations are weak? Are there failure modes you can characterize?
5. Scalability: How does the approach scale to scenarios with more modalities (>3) or when different extreme percentages of the test distribution have missing modalities (5% or 95% missing modalities)? Is this method better under certain instances compared to others

---

### Official Review · Reviewer_rsUS · 2025-10-31

**Soundness:** 2
**Presentation:** 3
**Contribution:** 2
**Rating:** 2
**Confidence:** 3

**Summary:**

The paper proposes SELFMASK, a cross-modal self-masking framework for multimodal learning under missing-modality conditions. The method learns adaptive (data-driven) masks rather than random masks and uses a cross-modal consistency objective: predicted representations for missing modalities should be semantically aligned and also help reconstruct observed modalities.

**Strengths:**

- **Problem relevance**. Handling missing modalities is practically important in clinical and multimedia settings, and the paper targets recognized gaps in robustness.
- **Method intuition**. Learning informative masks echoes advances in learned/curriculum masking for MAE-style objectives (e.g., AutoMAE, CL-MAE; also guidance via token-critic), and a consistency signal across modalities is conceptually appealing.

**Weaknesses:**

- **Poor Illustration**. The overall pipeline figure (Fig.2) is not self-explained and cannot capture the idea/pipeline well enough.
- **Selection bias from "complete-case" construction**. The paper extracts only fully paired samples and then simulates missingness. This is somewhat restrictive, while current literature on missing modalities target much more generic settings where missing modalities actually happen during both training and evaluation.
- **Incremental novelty**. Learned masking has been explored in several works (AutoMAE, CL-MAE, self-guided MAE), and cross-modal masked pretraining is not new (e.g., Multimodal MAE). The paper’s technical delta over these lines is not crisply articulated [1,2,3].

[1] Chen, Haijian, et al. "Improving masked autoencoders by learning where to mask." Chinese Conference on Pattern Recognition and Computer Vision (PRCV). Singapore: Springer Nature Singapore, 2023.

[2] Madan, Neelu, et al. "Cl-mae: Curriculum-learned masked autoencoders." Proceedings of the IEEE/CVF Winter Conference on Applications of Computer Vision. 2024.

[3] Shin, Jeongwoo, et al. "Self-guided masked autoencoder." Advances in Neural Information Processing Systems 37 (2024): 58929-58954.

**Questions:**

Please refer to Weaknesses.

---

### Official Review · Reviewer_EWBh · 2025-11-01

**Soundness:** 2
**Presentation:** 2
**Contribution:** 2
**Rating:** 4
**Confidence:** 4

**Summary:**

This paper proposes a cross-modal self-masking framework, SelfMask, for learning robust multimodal representations with missing modalities. The key idea is to leverage an adaptive masking strategy (learned via a mask predictor) and a cross-modal consistency loss to impute missing modality representations without ground truth. The method is evaluated on MIMIC-IV and CMU-MOSEI datasets, demonstrating improvements over several strong baselines under simulated missing-modality conditions.

**Strengths:**

+ The paper addresses the missing modality problem, which is a significant and practical barrier to deploying multimodal systems in real-world settings
+ The proposed framework is model-agnostic and can attach to standard encoders and a simple fusion module.
+ The cross-modal consistency loss provides a reasonable way to supervise imputation when ground truth for the missing modality is unavailable.

**Weaknesses:**

- The proposed "adaptive masking" strategy is repeatedly contrasted with "conventional random masking”. However, the ablation study in Table 4 only evaluates the different loss components. There is no experiment in the paper that compares the proposed adaptive masking ($h_{\omega}$) against a standard, computationally simpler baseline.
- More specifically, the evaluation is missing several strong but simpler control experiments that could strengthen the paper. For example: i) A simple modality dropout or zero-imputation baseline combined with a robust fusion module (no predictor). ii) A standard random masking strategy. iii) Simpler latent-space imputation methods (e.g., k-nearest neighbors) with a comparable parameter budget.
- The "cross-modal consistency" loss functions as a reconstruction constraint. However, there is limited analysis of what semantic properties are actually preserved in the imputed representations ($\hat{Z}^{(m)}$). An analysis of class-conditional fidelity or visualization of the imputed latent space would be beneficial.
- The complete framework, including the representation predictor ($g_{\phi}$) and mask predictor ($h_{\omega}$), is required during inference for missing modalities. The paper does not quantify the latency, FLOPs, or memory overhead.

**Questions:**

(1) Could you please provide the ablation study that compares your adaptive "mask prediction" ($h_{\omega}$) against a standard, computationally cheaper random masking strategy with the same masking ratios?

(2) How do you expect the proposed framework, particularly the adaptive masking and consistency loss, to scale to high-modality scenarios (e.g., 5 or 10 modalities)?

(3) Could you provide a more detailed analysis of the computational cost (parameters, training time, and inference-time FLOPs/latency) to better contextualize the performance-vs-complexity trade-off of SelfMask against the baselines?

(4) Can you deepen the discussion on the learned masks (e.g., via visualization or quantification) to help us understand what patterns or features the mask predictor ($h_{\omega}$) is learning to identify as "informative"?

---

### Official Review · Reviewer_V93W · 2025-11-03

**Soundness:** 2
**Presentation:** 2
**Contribution:** 2
**Rating:** 2
**Confidence:** 3

**Summary:**

This paper introduces a framework designed to learn robust multimodal representations in the presence of missing modalities. The method employs an adaptive masking strategy that learns which parts of observed modalities to mask and uses a cross-modal consistency loss to impute missing modality representations without ground truth. The approach is evaluated on two benchmarks under various missing-modality scenarios.

**Strengths:**

1. The paper is clearly and concisely structured, making it easy to follow.

2. The method is systematically evaluated on two diverse datasets under various missing-modality settings. An ablation study effectively demonstrates the contribution of each loss component.

**Weaknesses:**

1. Most building blocks exist (MAE/JEPA-inspired latent prediction, EMA teacher, Top-K masking). The novelty rests on how they’re assembled for missing-modality training.

2. The paper claims that learned adaptive masks outperform random masking, yet no quantitative analysis is provided. For example, do the adaptive masks consistently target tokens with higher semantic or task-relevant information?

3. The current evaluation uses controlled and fixed missing-modality patterns. In real-world datasets, missingness can be heterogeneous across samples (the specific missing modalities may vary). Could the authors comment on how well SelfMask is expected to perform under such more realistic and heterogeneous missingness patterns?

**Questions:**

1. The paper introduces zero-fill and random-fill as baselines but provides minimal implementation details. Moreover, some methods (e.g., SMIL, MedFuse) perform comparably or even worse than these simple imputations on certain metrics. Could the authors clarify why this might occur and what it implies about the inherent challenges of the task?

2. In Figure 3, the learned mask appears visually random. Could the authors provide a quantitative comparison or ablation showing that the learned mask indeed outperforms random masking? Can the authors elaborate on the semantic meaning or provide qualitative insights from Figure 3?

3. Do the authors think that strong pretrained encoders (e.g., SigLIP2) may overshadow the contribution of the masking mechanism?

4. There are minor typos (e.g., in Section 3.6: “our method achieves only a 7.9 percentage point drop…”). It is unclear where the “7.9” figure comes from. Please verify or clarify.

---

### Author Response · Authors · 2025-11-26
**General Response: Clarification of Novelty and Contributions**

We thank all reviewers for their thoughtful comments and constructive suggestions. Several reviews highlight both the importance of the missing-modality setting and the potential of representation-level cross-modal imputation, which is precisely the problem our work aims to address. We also appreciate the concrete experimental suggestions (e.g., additional analyses of the learned masks and alternative baselines) and are currently running these experiments; we will incorporate the new results and clarifications into the revised manuscript.

---

Several reviewers (V93W, rsUS, pQCE) raised concerns about incremental novelty, noting that our method builds on familiar ingredients such as MAE/JEPA-style latent prediction, EMA teachers, and Top-K masking. We fully agree that these primitives are not new by themselves. The contribution of **SelfMask** lies in how these ingredients are integrated into **a cross-modal self-masking framework** specifically tailored to missing-modality training across heterogeneous modalities and diverse, sample-wise heterogeneous missing configurations.

Concretely, our contribution has two main aspects:

1. **Cross-modal mask prediction for missing-modality robustness:**
 Prior learned-masking approaches (AutoMAE, CL-MAE, Self-guided MAE) learn where to mask within a single modality to improve reconstruction or representation quality of that same modality. In contrast, SelfMask’s mask predictor operates on **joint multimodal representations**, conditioned on both observed modalities and missing-modality tokens. The masks are learned to improve **prediction of missing-modality representations**, not just single-modality reconstruction. This cross-modal, prediction-aware masking differs from multimodal MAE variants such as M3AE and MultiMAE, as well as ActionMAE, which primarily rely on random or modality-local masking and do not adapt masks based on cross-modal interactions under missing-modality simulation.

2. **Cross-modal consistency for representation-level imputation without targets:**
In standard MAE/JEPA-style settings, the latent targets for masked tokens are always available. In the missing-modality regime, the target representation for an actually missing modality is fundamentally unobserved. SelfMask addresses this by jointly predicting representations for both observed and missing modalities and enforcing a **cross-modal consistency loss**, where the predicted missing-modality embeddings must be informative enough to reconstruct heavily masked representations of the observed modalities. This yields a multi-objective training scheme that explicitly couples (i) reconstruction of adaptively masked observed tokens and (ii) consistency-driven imputation of missing views, which is not present in cross-modal translation/alignment methods such as CMAT, which assume observed targets for the translated modality, nor in multimodal MAEs that simply extend random masking to multiple modalities.

Regarding concurrent work such as Robult, which maximizes redundancy via mutual information while preserving modality-specific information through unimodal latent reconstruction, SelfMask instead **reconstructs a modality’s representation from other modalities under explicit missing-modality simulation**. Conceptually, Robult is closer to shared/specific factor modeling (e.g., ShaSpec-style decompositions), whereas our framework targets representation-level imputation via adaptive cross-modal masking and consistency.

---

We will revise the manuscript to more clearly articulate how SelfMask differs from prior learned-masking, multimodal MAE, and representation-imputation approaches, and to emphasize that our main contribution lies in integrating cross-modal self-masking and consistency objectives for missing-modality training. We are grateful to the reviewers for pointing us to relevant and concurrent lines of work, and we will explicitly incorporate and contrast these references to better position the contribution and scope of our work.

Within the remaining review period, we are prioritizing the key experiments suggested by the reviewers (e.g., comparisons against random masking and stronger baselines, and additional analyses of the learned masks) and will update the manuscript accordingly to further address the concerns regarding novelty and empirical validation.

---

### Note · Program_Chairs · 2026-01-17
**Submission Desk Rejected by Program Chairs**

The following references in this submission do not refer to real documents and/or have major errors in bibliographic information:

     Kejing Yin, Peng Wang, Zhi Liu, and Shan Wang. Robult: An information-theoretic approach for robust multimodal learning with missing modalities. In International Conference on Learning Representations, 2023.
    Kejing Yin, Peng Wang, Zhi Liu, and Shan Wang. M3-jepa: A multimodal framework based on joint-embedding predictive architecture. arXiv preprint arXiv:2409.05929, 2024.
    Hritik Shah, Ziyu Shen, Zixian Zhao, Si Fu, Zhaoyang Zhang, D. Yu Wang, Fei Wang, and Jieping Ye. Muse: Multimodal self-supervised learning for clinical data. In International Conference on Learning Representations, 2023.